# Using Bee Products for the Prevention and Treatment of Oral Mucositis Induced by Cancer Treatment

**DOI:** 10.3390/molecules24173023

**Published:** 2019-08-21

**Authors:** Karsten Münstedt, Heidrun Männle

**Affiliations:** Department of Gynaecology, Ortenau Klinikum, 77654 Offenburg, Germany

**Keywords:** oral mucositis, chemotherapy, radiotherapy, radio-chemotherapy, honey, propolis, royal jelly

## Abstract

Oral mucositis is one of the most frequent complications after chemotherapy or radiotherapy or a combination of both. There is no standard therapy for its prevention or treatment. Considering that some bee products have been found to be of value in this situation, we decided to analyze the scientific literature on the subject. Scientific publications on bee products were identified by a literature search on Pubmed, Scopus and Google Scholar. There is a lot of evidence regarding the use of honey for oral mucositis due to chemotherapy or radiotherapy or a combination of both. Unfortunately, the quality of several meta-analyses on the topic is very low. There is some evidence on propolis, a little on royal jelly and none whatsoever on pollen and other bee products like apilarnil or bee venom. Bee products such as honey, propolis and royal jelly may be well suited to be integrated into a general concept for the prevention and treatment of oral mucositis which should also include other established concepts like oral care, oral cryotherapy, topical vitamin E and low-level-laser therapy. Bee products could become an integral part in the treatment of chemotherapy, radiotherapy and radio chemotherapy. High-quality meta-analyses and further studies, especially on the combinations of various strategies, are needed.

## 1. Introduction

Oral mucositis is one of the most common complications after chemotherapy or local radiotherapy (e.g., for head and neck cancer) or a combination of both. Its severity depends on the type and dose of chemotherapy and/or radiotherapy. Oral mucositis results in erythema and the ulceration of the non-keratinized mucosa, causing significant pain, compromised nutrition, prolonged hospitalization and bloodstream infections. Because of these symptoms, it can cause treatment delays and impact the delivery of optimal cancer treatment, thus affecting clinical remissions and the chances of a cure. 

The mechanisms of the development of oral mucositis are not fully understood, especially the specific downstream cellular events resulting in oral epithelial damage. With respect to chemotherapy, the cytotoxic agents will affect all rapidly dividing epithelium along the gastrointestinal tract including the mouth, larynx and pharynx via impaired DNA replication and repair, cell-cycle arrest, DNA damage and cell death. Furthermore, it has been suggested that oral microbiome communities contribute to the development of mucositis [1]. During radiotherapy, reactive oxygen species released from epithelial and endothelial cells will activate NF-кB and TNF (tumour necrotic factor) producing pro-inflammatory cytokines, influencing ceramide synthase production and causing DNA damage injury and cell death. Once again, bacteria are believed to influence these processes [2]. A detailed and comprehensive review of this topic was recently published by Bowen et al. [3]. 

In cases of a combination of chemotherapy and radiotherapy, it is believed that the two mechanisms come together, which aggravates the problem. Furthermore, several factors have been found to be associated with increased risk for oral mucositis. These include age (increased risk in very young and old age), gender (increased risk in females), oral health and hygiene, low salivary secretory function, genetic factors, body mass index (increased risk in malnourished individuals), poor renal function, smoking and previous cancer treatment [2].

Various methods have been suggested in order to prevent and treat oral mucositis. The German S3-Guideline for supportive care in cancer (a guideline following a systematic development) recommends standardized oral care consisting of a regular application of mouth rinses, dental hygiene (use of a soft toothbrush, use of dental floss), the avoidance of noxious substances (such as alcohol, tobacco, spicy and hot dishes or acidic food), a continuous control of lesions and pain, prophylactic measures by dentists, fluoridation for dental protection and clinical control and counseling during therapy [4]. 

However, standard oral care alone is not enough for patients receiving such treatments [5]. Unfortunately, there are few other options offered by conventional medicine. A systematic review found that the commonly used chlorhexidine was neither effective in reducing the severity of mucositis nor in preventing its incidence [6]. Concepts primarily come from complementary and integrative medicine. However, the interpretation of the results from various trials is conflicting. For example, one meta-analysis concluded that glutamine may reduce the risk and severity of mucositis during radiotherapy or chemotherapy in general [7], whereas The Mucositis Study Group of the Multinational Association of Supportive Care in Cancer presented a more differentiated conclusion and recommended against parenteral glutamine in hematopoietic stem cell transplantation patients and voted in favor of oral glutamine in head and neck cancer patients [8]. For many other vitamins, minerals and nutritional supplements, no positive recommendations were possible [8]. Low-level-laser therapy has proven its efficacy in a systematic review and meta-analysis [9] but oral cryotherapy is generally recommended, especially when patients receive 5-fluorouracil chemotherapy [4,10,11]. 

Little is known on what patients use except for one study from pediatric oncology. Black mulberr, carbonate and honey (11.6%) were the methods used most frequently [12]. The keratinocyte growth factor may be another interesting option for patients who are receiving either radiotherapy to the head and neck with cisplatin or fluorouracil or chemotherapy alone for mixed solid and hematological cancers [13]. 

Despite this, the past few years have seen the publication of a considerable number of clinical trials on the use of bee products for oral mucositis with interesting results. Therefore, this review is going to summarize the current evidence of those bee products.

## 2. Results

### 2.1. Chemotherapy-Induced Oral Mucositis and Honey

There is a systematic review on honey for the treatment of chemotherapy-induced mucositis in paediatric oncology patients [14]. It summarizes four papers on the topic and provides Grade C evidence that honey is effective as a preventative and therapeutic measure in paediatric oncology patients. However, two studies not only investigated pure honey. The study by Abdulrhman et al. analyzed honey as well as a combination of honey, olive oil, propolis and beeswax compared to a control group and found that both groups with honey were superior to the control [15]. In less severe cases, honey alone was the best alternative, whereas in more severe cases the combination of honey, olive oil, propolis and beeswax was superior. Another study compared sugar syrup, which was added with some betamethasone to a honey solution, which in turn was added to instant coffee powder and compared to a pure honey group [16]. Interestingly, the combination of honey and coffee was most likely to improve patients’ complaints. Additional recent studies confirm the positive effects of honey [17]. However, a trial comparing honey, vitamin E and chlorhexidine found that vitamin E was the best alternative [18]. The fact that topical vitamin E is another interesting option is confirmed in a meta-analysis [19]. 

As mentioned, the evidence for chemotherapy comes from studies in paediatric oncology. However, it seems reasonable to assume that chemotherapy-induced oral mucositis will not differ substantially between children and adults because the underlying pathomechanisms are alike. 

One very interesting study combined honey, tulsi (*Ocimum tenuiflorum*) and cryotherapy and found that the combination is superior to cryotherapy alone [20]. Unfortunately, the authors do not provide details on how the honey-tulsi-ice cubes were made. On one side, it appears that they used honey but on the other side they speak of honey-flavored ice cubes. Unfortunately, the authors Mishra and Nayak have not responded to our request to clarify the information in their paper. 

### 2.2. Radiotherapy-Induced (Radiation-Induced) Oral Mucositis

Patients with head and neck cancer mainly depend on radiation therapy in combination with chemotherapy, usually cis-platinum, especially when surgery cannot be done or was unable to remove all the cancer tissue with free margins. Several studies have addressed the question on whether honey may be a means to prevent radiotherapy-induced oral mucositis. These trials have been analyzed by various systematic reviews and even meta-analyses. An overview of trials is given in Table 1. As shown, all trials revealed positive results for honey. Manuka honey was not tested in this situation.

### 2.3. Honey and the Combination of Radiotherapy and Chemotherapy

Unfortunately, there are no trials which have specifically compared the effects of radiotherapy alone to the combination of radiotherapy and chemotherapy on oral mucositis with respect to the protective effects of honey. However, in the systematic review by Münstedt et al. there were nine trials on patients who received a combination of radiotherapy and chemotherapy [29]. Since conventional honey proved to be effective in this situation, it can be recommended here as well. A more recently published trial on the subject confirms the value of honey in this situation 30]. A detailed overview on these trials is given in Table 2. As shown, honey, but not manuka honey, is beneficial. All four trials using manuka honey failed to show an advantage for honey, whereas all trials using conventional honey led to positive results [30]. The reasons for this are unclear but may relate to the unusually high content of methylglyoxal in manuka honey which can be regarded as a cytotoxic agent. Another trial showed that a combination of glycerine plus honey was more effective than the standard treatment [31]. Similarly, a study analyzed the local application of Yashtimadhu (*Glycyrrhiza glabra*) powder and honey along with oral intake of Yashtimadhu Ghrita versus local application of the Yashtimadhu powder and honey versus local application of honey only or conventional modern medication [32]. It claimed to have found best results for the combination of Yashtimadhu and honey but the results are not convincing because patients were not equally distributed over the four treatment arms and drop-out rates left only nine patients in the honey arm [24]. However, Glycyrrhiza glabra seems to be an interesting option for the treatment of oral mucositis, as shown in another trial [33].

### 2.4. Working Mechanisms of Honey

Unfortunately, there are no studies which have investigated the working mechanisms of honey in oral mucositis. However, it has been shown that honey could influence aspects in the pathogenesis of oral mucositis. Since it was shown that pathogens, especially bacteria, could play a role in oral mucositis, the antibacterial properties of honey could be beneficial [43]. Since wounds due to radiotherapy have been found to be very similar to burns, honey’s antimicrobial and anti-inflammatory properties, its ability to autolytically debride and deodorize and its ability to stimulate tissue growth and to manage pain and minimize scarring are likely to be helpful as well [44,45]. 

### 2.5. Propolis and Radiotherapy

Evidence regarding the usefulness of propolis in radiotherapy dates back to 1989, when it was shown that both water and alcohol-based propolis extracts reduce radiotherapy-induced/radiation-induced mucositis and its associated swellings. Thus, the scheduled completions of radiotherapy were higher due to the intervention of propolis [46,47]. Another small, more recent, study on the topic determined the beneficial effects of propolis in comparison to control (water) [48]. Meanwhile, another study has confirmed that propolis efficiently prevents and heals radiation-induced mucositis [49]. As mentioned before, the study by Abdulrhman et al. analyzed a combination of honey, olive oil, propolis and beeswax and found that in more severe cases this combination was superior to honey and a control group [15]. This calls for further research on the benefit of propolis alone versus the benefit of a honey–propolis combination. 

It was also shown that propolis leads to less DNA damage in healthy cells which could explain part of its preventive effect [50]. Another mechanism could be its antibiotic effect against bacteria and viruses [51,52].

### 2.6. Propolis and Chemotherapy

The first investigation on the subject found no evidence for the benefit of propolis but found a 6% difference in favor of propolis [53]. This study only comprised 40 patients and the small number of patients may have been the reason why the level of significance was not reached. Further studies showed that propolis was superior to placebo or control groups [49,54,55,56].

### 2.7. Royal Jelly and Chemotherapy-Induced Oral Mucositis

After an in vivo animal trial showed wounds induced by chemotherapy healing faster with a royal jelly containing chitosan-sodiumalginate-film, clinical trials were initiated [57]. The first trial, comprising of 103 patients, investigated royal jelly in combination with antiphlogistic and antifungal treatments for the treatment of oral mucositis [58]. In comparison to controls and depending on the severity of mucositis, healing was significantly faster with royal jelly (up to 6.5 days). Another small trial of 13 patients also provided some evidence for its usefulness in this clinical setting [59]. 

### 2.8. Pollen and Other Bee Products 

There are no studies yet which have used bee pollen as a means to prevent or treat oral mucositis. However, studies on date palm pollen suggest a potential benefit [60]. There are no studies on other bee products such as apilarnil or bee venom on the subject.

## 3. Discussion

The presented data show that some bee products (honey, propolis, royal jelly and pollen) may be interesting candidates for the prevention and treatment of oral mucositis due to chemotherapy or radiotherapy or a combination of both. Clearly, the evidence regarding honey and radiation-induced oral mucositis is strongest [29]. The finding of the mentioned systematic review that conventional honey, but not manuka honey, seems to be a good treatment option cautions us to generalize the positive effects of all types of honeys and reminds us that all bee products do not have uniform properties. According to most studies patients should receive about 20 mL 15 min before and after radiotherapy and a third dose 6 h later. In the case of propolis, different plant sources as well as different extraction methods may result in different medicinal properties [61]. Thus, it seems important that the relevant substances and production methods of bee products in clinical trials are clearly specified. Furthermore, it is important to note that some trials used diluted bee products or combinations of bee products with other substances which might result in negative results [62]. Adverse interactions are possible. However, the efficacy of honey may be improved by the addition of effective substances or new ways of administration, e.g., honey cryotherapy.

For incomprehensible reasons, several meta-analyses had great methodological flaws and mixed up studies on radiotherapy- or chemotherapy-induced mucositis, or analysed studies were not only on bee products but plant extracts or only honey flavoured products [63,64,65,66,67,68]. The problems of these meta-analyses have been described in greater detail elsewhere [29,69]. If bee products should become part of the treatment concept for oral mucositis, it is important that the systematic reviews and meta-analyses are done properly. 

Apart from the analyses on efficacy regarding the prevention and treatment of oral mucositis, it must be noted that propolis is a substance with many pharmacologically active molecules and that there is a chance that propolis may interfere with chemotherapy, e.g., bleomycin [70,71]. Such drug interactions could lead to the impaired efficacy of chemotherapy, which must be avoided. In this respect, it must be ascertained whether the efficacy of the basic treatment is compromised or not. This should also be a matter of future investigations. 

Regarding the potential use of bee products in this area and acknowledging that the complementary and integrative measures for the prophylaxis and treatment of oral mucositis should be effective, safe, cost effective and readily available, algorithms can be established. Consideration of the named factors seems to be appropriate in order to provide the best care to the patient. Since direct comparisons between the various treatment options are scarce, the strength of these suggestions is still limited. Based on the findings of this review and the above-mentioned concept, the following approaches can be suggested (Table 3, Table 4, Table 5 and Table 6). The number of pluses indicates the strength of evidence (+++ = very good evidence; ++ = good evidence; + = some evidence).

While cryotherapy, oral care, topical vitamin E and low-level-laser therapy can be used without risk, it must be remembered that royal jelly and propolis have some allergic potential, so decisions regarding their use must be made very carefully and with specific patient information.

The use of honey ice cubes, as mentioned by Mishra and Nayak, deserves further attention because it combines two interesting treatment options [20]. Furthermore, it seems important that not only single treatments, but sequential treatment concepts are investigated, i.e., oral care as a basis, cyrotherapy during the infusion of cytotoxic drugs, honey during the days after chemotherapy and during radiotherapy instead of honey alone. Further high-quality research is necessary to further improve the standards of oncological care.

## 4. Materials and Methods

Scientific publications on bee products were identified by a literature search on Pubmed, Scopus and Google Scholar. All publications which seemed appropriate for this review were read in full-text form.

## 5. Conclusions

The use of the bee products honey and propolis seems to be very appropriate for the prophylaxis and treatment of oral mucositis induced by radiotherapy, chemotherapy and a combination of both. Further studies on co-administered methods and the best sequences are necessary. 

## Figures and Tables

**Table 1 molecules-24-03023-t001:** Summary of randomized trials on honey and radiation therapy. All studies had positive results for honey.

First Author/Year/Jadad Score	Sample; Intervention and Control Group	Endpoints (OM = Oral Mucositis; QoL = Quality of Life)	Main Results (OM = Oral Mucositis; QoL = Quality of Life)
Charalambous 2018 [21] 3	86 patients Group 1: 43 patients with diluted thyme honey (20 mL of thyme honey in 100 mL water making gargles in the oral cavity—¼ h before and after radiotherapy and 6 h later) Group 2: 43 patients with saline 0.9%	OM grade Weight loss Oral problems (i.e., swallowing, drinking, eating, mouth and throat pain) QoL	Lower grades of OM Better maintenance of body weight Improvement in global health Better QoL in the honey group No study discontinuation because of honey
Amanat 2017 [22] 3	82 patients Group 1: 41 patients with ziziphus honey (20 mL—¼ h before and after the radiotherapy) Group 2: 25 patients with saline 0.9%	OM grade	Lower grades of OM in the honey group No study discontinuation because of honey
Bahramnezhad 2015 [23] 0	105 patients Group 1: 35 patients with diluted polyfloral honey (50 mL honey and 25 mL water—administration schedule not given) Group 2: 35 patients with water Group 3: 35 patients with water chamomile	OM grade	Lowest grades of OM in the honey group No study discontinuation because of honey
Alvi 2013 [24] 2	60 patients Group 1: 30 patients with polyfloral honey (20 mL—¼ h before and after radiotherapy and 6 h later) Group 2: 30 patients with saline 0.9%	OM grade Weight loss Treatment discontinuation	Lower grades of OM Better maintenance of body weight No study discontinuation because of honey in the honey group No discontinuation of radiotherapy in the honey group (0 vs. 3 in the control group)
Jayachandran [25] 2012 2	60 patients Group 1: 20 patients with polyfloral honey (20 mL—¼ h before and after radiotherapy and 6 h later) Group 2: 20 patients with benzydamine hydrochloride Group 3: 20 patients with saline 0.9%	OM grade Onset of OM Recovery after end of therapy	Later onset of OM Lower grades of OM during radiotherapy Faster recovery from OM after the end of radiotherapy in the honey group No study discontinuation because of honey
Khanal 2010 [26] 2	40 patients Group 1: 20 patients with polyfloral honey (20 mL—¼ h before and after radiotherapy and before going to bed) Group 2: 20 patients with lignocaine Gel	OM grade Pain associated with OM	Lower grades of OM Less pain in the honey group No study discontinuation because of honey
Motallebnejad 2008 [27] 2	40 patients Group 1: 20 patients with thyme and astragal honey (20 mL—¼ h before and after radiotherapy and 6 h later) Group 2: 20 patients with saline 0.9%	OM grade Weight loss	Lower grades of OM Better maintenance of body weight in the honey group 4 patients in the honey group with Grade 0 OM discontinued treatment
Biswal 2003 [28] 2	40 patients Group 1: 20 patients with tea plant honey (20 mL—¼ h before and after radiotherapy and 6 h later) Group 2: 20 patients with saline 0.9%	OM grade Weight loss Treatment discontinuation Duration of mucositis	Lower percentage of patients with grade 3/4 mucositis Better maintenance of body weight in the honey group No discontinuation of radiotherapy in the honey group (0% vs. 20% in the control group) Duration of mucositis = n.s. No report of study discontinuation because of honey

**Table 2 molecules-24-03023-t002:** Summary of randomized trials on honey and the combination of chemotherapy and radiation therapy. (Studies highlighted in green mean positive results for honey, studies highlighted in red mean negative results for honey.) Note: All studies on manuka honey had negative results.

First Author /Year/Jadad Score	Sample; Intervention and Control Group	Endpoints (OM = Oral Mucositis; QoL = Quality of Life)	Main Results (OM = Oral Mucositis; QoL= Quality of Life)
Howlader 2019 [30] 1	40 patients Group 1: 20 patients with 20 mL polyfloral honey ¼ h prior to radiation, and ¼ h and 6 h after radiation Group 2: 20 patients rinse with saline ¼ h before and ¼ h after radiation exposure	OM grade QoL	Lower grades of OM at the end of treatment Better quality of life Fewer treatment interruptions in the honey group
Rao 2017 [34] 2	50 patients Group 1: 25 patients with polyfloral honey (exact quantity not given—1 h prior to radiation, and 2 and 6 h after radiation) Group 2: 25 patients with povidone-iodine	OM grade Treatment interruptions Tumour response	Lower grades of OM Better maintenance of body weight Fewer treatment interruptions in the honey group Honey has no effect on tumour response No study discontinuation because of honey
Fogh 2017 [35] 3	163 patients Group 1: 53 patients with supportive careGroup 2: 54 patients with manuka honey (10 mL—lozenges 4 times per day, over a period of 12 h daily) Group 3: 56 patients with lozenge manuka honey (2 lozenges 4 timesper day, over a period of 12 h daily)	Pain on swallowing QoL Secondary endpoints Pain over time Opioid use Clinically graded and patient-reported adverse events Weight loss Dysphagia Nutritional status,	No significant difference in the primary endpoint No differences in any of the secondary endpoints except for opioid use More patients on the supportive care arm took opioids
Jayalekshmi 2016 [36] 3	28 patients Group 1: 14 patients with polyfloral honey (15 mL—¼ h before and after radiotherapy and 6 h later) Group 2: 14 patients with water	OM grade	Lower grades of OM in the honey group No study discontinuation because of honey
Samdariya2015 [37] 3	78 patients Group 1: 40 patients with polyfloral honey (20 mL—¼ h before and after radiotherapy and 6 h later) and salt-soda and benzydamine gargle Group 2: 38 patients with salt-soda and benzydamine gargle	Pain	Less pain in the honey group No study discontinuation because of honey
Hawley 2014 [38] 5	106 patients Group 1: 54 patients with manuka honey (5-mL -four times a day after radiotherapy and after meals) Group 2: 52 patients with placebo gel	OM grade	No significant difference in the primary endpoint High dropout rates 57% in the honey group vs. 52% in the control group n.s.
Maiti 2012 [39] 1	55 patients Group 1: 28 patients with polyfloral honey (20 mL—¼ h before and after radiotherapy and 6 h later) Group 2: 27 patients no additional treatment	OM grade Weight loss	Lower grades of OM, Better maintenance of body weight in the honey group No study discontinuation because of honey
Bardy 2012 [40] 5	131 patients Group 1: 67 patients with manuka honey (20 mL—4 times a day) Group 2: 64 patients with golden syrup	OM grade OM duration Secondary aims assessment of microbiological flora in the mouth, requirements for antimicrobial drugs and analgesia Weight loss Need for tube feeding	No significant differences in the primary and secondary endpoints No differences in patients’ compliance
Parsons 2012 [41] 1	28 patients Group 1: 6 patients with manuka honey (20 mL—¼ h before and after radiotherapy and 6 h later) Group 2: 12 patients with diluted manuka honey (10 mL in 30 mL of water—¼ h before and after radiotherapy and 6 h later) Group 3: 10 patients receiving standard care	OM grade Weight loss QoL	No significant difference regarding OM grade Better maintenance of body weight in the honey group Diluted manuka honey increased overall QoL in the radiotherapy group but not in group of radiotherapy in combination with chemotherapy 6/6 patients in the honey group, 2/12 in the diluted honey group and 2/10 in the control group withdrew from the study because of pain and nausea
Rashad 2009 [42] 2	40 patients Group 1: 20 patients with clover honey (20 mL—¼ h before and after radiotherapy and 6 h later) Group 2: 20 patients receiving standard care	OM grade assessment of microbiological flora in the mouth	Lower grades of OM Lower rates of pathogenic bacteria and fungi in the honey group

**Table 3 molecules-24-03023-t003:** Suggestions regarding prophylaxis of chemotherapy-induced oral mucositis.

Method	Effectiveness	Safety
Oral care	+++	+++
Oral cryotherapy	+++	+++
Honey	++	++
Topical vitamin E	+++	+++
Low-level-laser therapy	+++	+++
Glutamine	++	++
Propolis	++	++

**Table 4 molecules-24-03023-t004:** Suggestions regarding treatment of chemotherapy-induced oral mucositis.

Method	Effectiveness	Safety
Oral care	+	+
Honey	++	++
Topical vitamin E	+++	+++
Low-level-laser therapy	+++	+++
Royal jelly	+	+

**Table 5 molecules-24-03023-t005:** Suggestions regarding prophylaxis of radiation-induced oral mucositis.

Method	Effectiveness	Safety
Oral care	+++	+++
Honey	+++	+++
Topical vitamin E	+++	+++
Low-level-laser therapy	++	++
Glutamine	++	++

**Table 6 molecules-24-03023-t006:** Suggestions regarding treatment of radiation-induced oral mucositis.

Method	Effectiveness	Safety
Oral care	+++	+++
Honey	+++	+++
Topical vitamin E	+++	+++
Low-level-laser therapy	++	++
Royal jelly	+	+

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
