# Peer review of "Using Bee Products for the Prevention and Treatment of Oral Mucositis Induced by Cancer Treatment"

_molecules, 2019, doi:10.3390/molecules24173023_

Round 1
Reviewer 1 Report
In this review, the authors dedicated a carefully reading of specialized literature. However, I did not found clients for this paper. The article is very superficial for a health expert and does not help a layperson much. Some parts of the text lack more secure information, for instance: the honey as the propolis have different origins. I do not believe that all kind of honey has the same composition; some of them could also be dangerous for the patient, or not indicated. For security, I believe that the authors should investigate the origins of the kinds of honey to approve or not for mucositis therapy.
Author Response
In this review, the authors dedicated a carefully reading of specialized literature. However, I did not found clients for this paper. The article is very superficial for a health expert and does not help a layperson much. Some parts of the text lack more secure information, for instance: the honey as the propolis have different origins. I do not believe that all kind of honey has the same composition; some of them could also be dangerous for the patient, or not indicated. For security, I believe that the authors should investigate the origins of the kinds of honey to approve or not for mucositis therapy.
Response 1:
We thank the reviewer. However it seems that the reviewer is not very familiar with the English language. We do not understand the sentence “However, I did not found clients for this paper.” We do not want to sell this paper and this was never our intention.
The reviewer also states “The article is very superficial for a health expert”. We completely disagree with this considering the fact that the literature on the subject does not even differentiate between oral mucositis due to chemotherapy o0r radiotherapy or a combination of both which we believe to believe to be essential. We never intended to write this article for lay persons.
We also do not understand the sentence “Some parts of the text lack more secure information, for instance: the honey as the propolis have different origins.” Both authors are beekeepers. Prof Münstedt now for more than 40 years. He is able to clearly distinguish between propolis and honey. Again – this article is for people with some general knowledge and we believe that is not necessary do explain the differences between propolis and honey.
The reviewer states: “I do not believe that all kind of honey has the same composition; some of them could also be dangerous for the patient, or not indicated. For security, I believe that the authors should investigate the origins of the kinds of honey to approve or not for mucositis therapy.” It ist stated in the manuscript: “As shown, honey, but not manuka honey, is beneficial. All four trials using manuka honey failed to show an advantage for honey, whereas all trials using conventional honey led to positive results [21].“ This statement makes it clear that there are differences between various honeys. We agree that it would be interesting to test various kinds of honeys against each other in clinical trials but this has not been done yet. We regret that we cannot provide information on this topic but such information is not available at present.
In summary, we found that the comments of this reviewer do not require any modification of the manuscript.
Reviewer 2 Report
Please, correct or improve in the text:
Line 113: what is 'tulsi'? Please explain in the text.
Lines 117&118: If the authors you mention haven't responded 'yet', then this statement isn't appropriate at this point. Please remove this statement unless you have some new information at the moment.
Lines 135-137: This statement is confusing. Please rewrite it to be more understandable.
Lines 146-147 and 153: 'Glycyrrhiza glabra' should be in italics
Lines 160-161: 'there is no detailed work on the working mechanisms of honey' – please rewrite this sentence in a better way. Explain the modes of action in wound healing therapy with honey (diabetics, injuries, burns etc.) reviewed by Stephen-Haynes (Wounds UK, 2011, Vol 7, No 1) to give to the reader a better overview of the value and importance of honey in medicine through the centuries.
Line 201: this statement is contradictory. If there are some studies on apilarnil and bee venom, why there is no literature presented?
Line 218: ‘by the addition of other substances’ – what kind? Natural or synthetic?
Lines 223-224: ‘The problems of this studies have been described elsewhere’ – What problems? Please explain and rewrite.
Lines 254-255: you mentioned this source above and found that there is no clear information in the same literature on what exactly are ‘honey-tulsi-ice cubes’ – actual honey or just a flavour? In some other literature on ice cubes and oral mucositis it is found that cryotherapy only helps a bit to mitigate the consequences of radiotherapy. First it must be clear about the flavoured ice cubes – is there an actual honey (and what kind) present and in what concentration etc.
Line 267: ‘seem to be very appropriate’ – are honey and propolis appropriate or are they not? Please be more specific.
Author Response
Line 113: what is 'tulsi'? Please explain in the text.
Response 1: We added the official name for tulsi ((Ocimum tenuiflorum) to the manuscript.
Lines 117&118: If the authors you mention haven't responded 'yet', then this statement isn't appropriate at this point. Please remove this statement unless you have some new information at the moment.
Response 2: The referred authors also did not answer our question in the meantime. We removed the word “yet”.
Lines 135-137: This statement is confusing. Please rewrite it to be more understandable.
2.3. Honey and the combination of radiotherapy and chemotherapy
Response 3: We have modified the sentence as requested.
Lines 146-147 and 153: 'Glycyrrhiza glabra' should be in italics
Response 4: Done as requested.
Lines 160-161: 'there is no detailed work on the working mechanisms of honey' – please rewrite this sentence in a better way. Explain the modes of action in wound healing therapy with honey (diabetics, injuries, burns etc.) reviewed by Stephen-Haynes (Wounds UK, 2011, Vol 7, No 1) to give to the reader a better overview of the value and importance of honey in medicine through the centuries.
Response 5: Done as requested. The suggested paper and its main messages are included in the revised version.
Line 201: this statement is contradictory. If there are some studies on apilarnil and bee venom, why there is no literature presented?
Response 5: We understand the reviewer’s problem. The sentences lacks a “no”
Line 218: ‘by the addition of other substances’ – what kind? Natural or synthetic?
Response 6: We have modified this sentence.
Lines 223-224: ‘The problems of this studies have been described elsewhere’ – What problems? Please explain and rewrite.
Response 7: We understand the reviewer’s problem and replaced studies with meta-analyses.
Lines 254-255: you mentioned this source above and found that there is no clear information in the same literature on what exactly are ‘honey-tulsi-ice cubes’ – actual honey or just a flavour? In some other literature on ice cubes and oral mucositis it is found that cryotherapy only helps a bit to mitigate the consequences of radiotherapy. First it must be clear about the flavoured ice cubes – is there an actual honey (and what kind) present and in what concentration etc.
Response 8: We understand the reviewer’s problem. We did not get an answer from the authors of the referred paper. We have done some experimentation on frozen honey and we believe that this is not very pleasant. So we believe that these are only honey flavored ice cubes.
Line 267: ‘seem to be very appropriate’ – are honey and propolis appropriate or are they not? Please be more specific.
Response 9: Done as requested
Reviewer 3 Report
Thank you for the opportunity to review this manuscript by Munstedt and Mannle. It provides a thorough overview of the current level of evidence supporting the use of bee products for the management of OM. It is clear that the authors are on top of the current literature on this topic, with appropriate reference to very recent publications from the MASCC/ISOO mucositis study group. I have a few comments for consideration:
Is this a review or systematic review? It is unclear to the reader and should be clearly stated. Given the presentation of the review with "results" and "discussion", I would consider it to be a systematic review. As such, I would recommend that the manuscript be updated to reflect the core processes of a systematic review, i.e. more detailed search strategy to be described in methods (keywords, inclusion/exclusion criteria, number of papers identified and excluded- and for what reasons).
The tables would benefit from a comment regarding the quality of the evidence, or some description of any study limitations. perhaps consider using the bias analysis approach typically used in a meta-analysis.
Authors mention briefly the potential mechanisms by which honey may be effective. Whilst I acknowledge there is limited data in terms of OM, description of the general mechanisms of honey would be beneficial to readers unfamiliar with this product. e.g. antimicrobial properties - what other studies have investigated this, what results have been shown? Does honey exert anti-inflammatory properties, how much is known about this mechanism?
Authors describe that some meta-analyses on this topic are flawed - more description on this would be beneficial to the reader. As it stands, the reader must find the cited study. A brief description of the limitations/flaws and how they could be improved should be included in the current manuscript.
Discussion of the potential impact of honey on chemoefficacy was a good inclusion by the authors. I think this section could be enhanced to describe the current level of evidence on this topic, i.e. what studies have shown bleomycin affects chemotherapy efficacy? By what mechanism? How could this be avoided? Is this the case for topical administration, or systematic? Please expand.
The recommendations provided by the authors are, in my opinion, heading into dangerous territory. The MASCC/ISOO Mucositis Study Group employs strict assessment criteria to provide evidence based clinical practice guidelines. These decisions are made by a large group of international experts across a variety of disciplines. Reaching a consensus takes months of careful analysis of the current level of evidence. In this manuscript, we have two people making recommendations with little to no description on how these recommendations were made, or the processes used to make these decisions. As such, whilst I commend the authors' intent on providing some clear structure and guidance to how OM is managed, I fear that these recommendations do not have the appropriate methods to support them. This is particularly difficult to justify with MASCC/ISOO currently unable to provide any recommendation regarding honey for OM. I therefore recommend that the authors consider removing these recommendations from the review. A summary table could still be included which synthesises the literature/evidence, but the term "recommendation" should be removed.
Thank you again for providing me the opportunity to review this manuscript. I look forward to seeing the revised copy.
Author Response
Thank you for the opportunity to review this manuscript by Munstedt and Mannle. It provides a thorough overview of the current level of evidence supporting the use of bee products for the management of OM. It is clear that the authors are on top of the current literature on this topic, with appropriate reference to very recent publications from the MASCC/ISOO mucositis study group. I have a few comments for consideration:
Is this a review or systematic review? It is unclear to the reader and should be clearly stated. Given the presentation of the review with "results" and "discussion", I would consider it to be a systematic review. As such, I would recommend that the manuscript be updated to reflect the core processes of a systematic review, i.e. more detailed search strategy to be described in methods (keywords, inclusion/exclusion criteria, number of papers identified and excluded- and for what reasons).
Response 1: Clearly, we have done this for honey and radiation-induced oral mucositis (Münstedt, K.; Momm, F.; Hübner, J. Honey in the management of side effects of radiotherapy- or radio/chemotherapy-induced oral mucositis. A systematic review. Complement. Ther. Clin. Pract. 2019, 34, 145-152. doi: 10.1016/j.ctcp.2018.11.016). But it would be too demanding to do a systematic review on all bee products and the various scenarios in which oral mucositis can occur. This article is meant to be a review as stated.
The tables would benefit from a comment regarding the quality of the evidence, or some description of any study limitations. perhaps consider using the bias analysis approach typically used in a meta-analysis.
Response 2: We added the Jadad-Scores to all studies.
Authors mention briefly the potential mechanisms by which honey may be effective. Whilst I acknowledge there is limited data in terms of OM, description of the general mechanisms of honey would be beneficial to readers unfamiliar with this product. e.g. antimicrobial properties - what other studies have investigated this, what results have been shown? Does honey exert anti-inflammatory properties, how much is known about this mechanism?
Response 3: The revised version includes the paper “Stephen Haynes, J.; Callaghan R. Properties of honey: its mode of action and clinical outcomes. Wounds UK 2011, 50-57” and mentions the various mechanisms for honey for burns in detail.
Authors describe that some meta-analyses on this topic are flawed - more description on this would be beneficial to the reader. As it stands, the reader must find the cited study. A brief description of the limitations/flaws and how they could be improved should be included in the current manuscript.
Response 4: We understand that the statement is misleading. But as stated in the text the meta-analyses “mixed up studies on radiotherapy or chemotherapy-induced mucositis or analysed studies were not only on bee products but plant extracts or only honey flavoured products. We take the most recent meta-analysis by Liu et al . as an example. It titles “Prophylactic and therapeutic effects of honey on radiochemotherapy-induced mucositis: a meta-analysis of randomized controlled trials”. We believe that the reader expects a review on honey and radiochemotherapy induced mucositis which means an analysis of patients who have received both radiotherapy and chemotherapy. Liu et al. 2019 claim to have analyzed 19 trials on honey for radiochemotherapy induced mucositis. We went through the various trials listed in the publication. Notably we found 13 trials which should not have been included in such an analysis. This means that 68 % of the trials should not have been included.
|
Study |
Reason why study should not have been included |
|
Abdulrhman et al. 2012 |
Patients did not receive radiotherapy |
|
Al Jaouni et al. 2017 |
Patients did not receive radiotherapy in the area of the head and neck |
|
Amanat et al. 2017 |
Patients did not receive chemotherapy |
|
Bansal et al. 2017 |
Patients did not receive honey but a mixture of glycerin and honey |
|
Biswal et al. 2003 |
Patients did not receive chemotherapy |
|
Charalambous et al. 2018 |
Patients did not receive chemotherapy |
|
Jayachandran et al. 2012 |
Patients did not receive chemotherapy |
|
Khanal et al. 2010 |
Patients did not receive chemotherapy |
|
Khanjani Pour-Fard-Pachekenari et al. 2018 |
Patients did not receive radiotherapy |
|
Mishra et al. 2017 |
Patients did not receive radiotherapy. The trial is on cryotherapy and it is unclear from the study whether the “flavoured” ice cubes are made entirely of honey or only to a certain extent. |
|
Motallebnejad et al. 2008 |
Patients did not receive chemotherapy |
|
Raeessi et al. 2014 |
Patients did not receive radiotherapy and some received a combination of honey and coffee |
|
Singh et al. 2018 |
Patients did not receive radiotherapy |
On the other hand the meta-analysis missed the studies by Howlader et al. 2019, Maiti et al. 2012, and Parsons et al. 2012.
We believe that such detailed analyses on all meta-analyses will distract the reader from our main topic. We have modified the sentence however.
Discussion of the potential impact of honey on chemoefficacy was a good inclusion by the authors. I think this section could be enhanced to describe the current level of evidence on this topic, i.e. what studies have shown bleomycin affects chemotherapy efficacy? By what mechanism? How could this be avoided? Is this the case for topical administration, or systematic? Please expand.
Response 5: There some misunderstanding. It is stated “it must be noted that propolis is a substance with many pharmacologically active molecules and that there is a chance that propolis may interfere with chemotherapy, e.g. bleomycin [52,53].” So the statement only refers to propolis and not to honey. We believe that it is clear from the titles of the cited references that there are preclinical studies only. We do not like to follow the advice of the reviewer to expand on this. We believe the topic is worth mentioning but there is not enough clinical evidence to give further details.
The recommendations provided by the authors are, in my opinion, heading into dangerous territory. The MASCC/ISOO Mucositis Study Group employs strict assessment criteria to provide evidence based clinical practice guidelines. These decisions are made by a large group of international experts across a variety of disciplines. Reaching a consensus takes months of careful analysis of the current level of evidence. In this manuscript, we have two people making recommendations with little to no description on how these recommendations were made, or the processes used to make these decisions. As such, whilst I commend the authors' intent on providing some clear structure and guidance to how OM is managed, I fear that these recommendations do not have the appropriate methods to support them. This is particularly difficult to justify with MASCC/ISOO currently unable to provide any recommendation regarding honey for OM. I therefore recommend that the authors consider removing these recommendations from the review. A summary table could still be included which synthesises the literature/evidence, but the term "recommendation" should be removed.
Response 6: We disagree with the reviewer. We did not make recommendations. It is stated in the manuscript: “Based on the findings of this review and the above-mentioned concept, the following approaches can be suggested.” Thus we do not make recommendations but suggestions. But we understand that giving a grade of recommendation can lead to this misunderstanding. We believe that replacing the term a grade of recommendation with grade of suggestion is not good either.
Thank you again for providing me the opportunity to review this manuscript. I look forward to seeing the revised copy.
Round 2
Reviewer 1 Report
To the Authors
I accepted the modifications of the text.
Author Response
I accepted the modifications of the text.
Response: We thank the reviewer
Reviewer 3 Report
Thank you for the changes and clarifications made throughout the manuscript. Based on the review comments from both R1 and R2, the manuscript is clearer and more comprehensive. While these changes have contributed positively to the overall quality of the manuscript, is still have significant concern over the inclusion of recommendation regarding the efficacy and safety of certain interventions (glutamine, vit E), which are in stark contrast to those made by MASCC.
Lalla et al., 2014: "The evidence reviewed supported the continuation of a recommendation against the use of intravenous glutamine for the prevention of oral mucositis in patients receiving high-dose chemotherapy for HSCT (Table 4). Due to inadequate and/or conflicting evidence, no guideline was possible in relation to other agents of natural origin reviewed, including glutamine in other treatment settings, the antioxidants vitamin A and E, honey, aloe vera, cha- momile, Kamillosan, Chinese herbals, indigowood root, manuka and kanuka oils, oral gel wafers, Rhodiola algida, traumeel S, and Wobe-Mugos E.2
This has been largely upheld in the 2019 update https://rdcu.be/bJmyA with glutamine only achieving a suggestion for prophylatic efficacy in H&N cancer patients. Additionally, vitamin E was unable to achieve any guideline due to insufficient evidence. How do the authors justify their "two smiley face" recommendation in contrast to the lack of guideline possible as per the MASCC guidelines?
I strongly advise that the authors reconsider the inclusion of their recommendation. Simply rephrasing this to an "evaluation of the strength of evidence" would be a simple method of overcoming this.
Author Response
Lalla et al., 2014: "The evidence reviewed supported the continuation of a recommendation against the use of intravenous glutamine for the prevention of oral mucositis in patients receiving high-dose chemotherapy for HSCT (Table 4). Due to inadequate and/or conflicting evidence, no guideline was possible in relation to other agents of natural origin reviewed, including glutamine in other treatment settings, the antioxidants vitamin A and E, honey, aloe vera, cha- momile, Kamillosan, Chinese herbals, indigowood root, manuka and kanuka oils, oral gel wafers, Rhodiola algida, traumeel S, and Wobe-Mugos E.2
This has been largely upheld in the 2019 update https://rdcu.be/bJmyA with glutamine only achieving a suggestion for prophylatic efficacy in H&N cancer patients. Additionally, vitamin E was unable to achieve any guideline due to insufficient evidence. How do the authors justify their "two smiley face" recommendation in contrast to the lack of guideline possible as per the MASCC guidelines?
Answer: As mentioned in line 110, a meta-analysis confirmed the positive effect of topical vitamin E. We inserted the word “topical” in the referred line to make this more clear.
Chaitanya NC et al. Role of Vitamin E and Vitamin A in Oral Mucositis Induced by Cancer Chemo/Radiotherapy- A Meta-analysis. J Clin Diagn Res. 2017 May;11(5):ZE06-ZE09. doi: 10.7860/JCDR/2017/26845.9905.
We have no idea why the MASCC chose to ignore this meta-analysis.
I strongly advise that the authors reconsider the inclusion of their recommendation. Simply rephrasing this to an "evaluation of the strength of evidence" would be a simple method of overcoming this.
Answer: Done as requested